# SVD-Softmax: Fast Softmax Approximation on Large Vocabulary Neural Networks

**Kyuhong Shim, Minjae Lee, Iksoo Choi, Yoonho Boo, Wonyong Sung**
Department of Electrical and Computer Engineering
Seoul National University, Seoul, Korea
skhu20@snu.ac.kr, {mjlee, ischoi, yhboo}@dsp.snu.ac.kr, wysung@snu.ac.kr

## Abstract

We propose a fast approximation method of a softmax function with a very large vocabulary using singular value decomposition (SVD). SVD-softmax targets fast and accurate probability estimation of the topmost probable words during inference of neural network language models. The proposed method transforms the weight matrix used in the calculation of the output vector by using SVD. The approximate probability of each word can be estimated with only a small part of the weight matrix by using a few large singular values and the corresponding elements for most of the words. We applied the technique to language modeling and neural machine translation and present a guideline for good approximation. The algorithm requires only approximately 20% of arithmetic operations for an 800K vocabulary case and shows more than a three-fold speedup on a GPU.

## 1 Introduction

Neural networks have shown impressive results for language modeling [1–3]. Neural network-based language models (LMs) estimate the likelihood of a word sequence by predicting the next word $w_{t+1}$ by previous words $w_{1:t}$. Word probabilities for every step are acquired by matrix multiplication and a softmax function. Likelihood evaluation by an LM is necessary for various tasks, such as speech recognition [4, 5], machine translation, or natural language parsing and tagging. However, executing an LM with a large vocabulary size is computationally challenging because of the softmax normalization. Softmax computation needs to access every word to compute the normalization factor $Z$, where $softmax(z_k) = \exp(z_k)/\sum_V \exp(z_i) = \exp(z_k)/Z$. $V$ indicates the vocabulary size of the dataset. We refer the conventional softmax algorithm as the "full-softmax."

The computational requirement of the softmax function frequently dominates the complexity of neural network LMs. For example, a Long Short-Term Memory (LSTM) [6] RNN with four layers of 2K hidden units requires roughly 128M multiply-add operations for one inference. If the LM supports an 800K vocabulary, the evaluation of the output probability computation with softmax normalization alone demands approximately 1,600M multiply-add operations, far exceeding that of the RNN core itself.

Although we should compute the output vector of all words to evaluate the denominator of the softmax function, few applications require the probability of every word. For example, if an LM is used for rescoring purposes as in [7], only the probabilities of one or a few given words are needed. Further, for applications employing beam search, the most probable top-5 or top-10 values are usually required. In speech recognition, since many states need to be pruned for efficient implementations, it is not demanded to consider the probabilities of all the words. Thus, we formulate our goal: to obtain accurate top-K word probabilities with considerably less computation for LM evaluation, where the K considered is from 1 to 500.

In this paper, we present a fast softmax approximation for LMs, which does not involve alternative neural network architectures or additional loss during training. Our method can be directly applied to full-softmax, regardless of how it is trained. This method is different from those proposed in other papers, in that it is aimed to reduce the evaluation complexity, not to minimize the training time or to improve the performance.

The proposed technique is based on singular value decomposition (SVD) [8] of the softmax weight matrix. Experimental results show that the proposed algorithm provides both fast and accurate evaluation of the most probable top-K word probabilities.

The contributions of this paper are as follows.

- We propose a fast and accurate softmax approximation, SVD-softmax, applied for calculating the top-K word probabilities.
- We provide a quantitative analysis of SVD-softmax with three different datasets and two different tasks.
- We show through experimental results that the normalization term of softmax can be approximated fairly accurately by computing only a fraction of the full weight matrix.

This paper is organized as follows. In Section 2, we review related studies and compare them to our study. We introduce SVD-softmax in Section 3. In Section 4, we provide experimental results. In Section 5, we discuss more details about the proposed algorithm. Section 6 concludes the paper.

## 2 Related work

Many methods have been developed to reduce the computational burden of the softmax function. The most successful approaches include sampling-based softmax approximation, hierarchical softmax architecture, and self-normalization techniques. Some of these support very efficient training. However, the methods listed below must search the entire vocabulary to find the top-K words.

**Sampling-based approximations** choose a small subset of possible outputs and train only with those. Importance sampling (IS) [9], noise contrastive estimation (NCE) [10], negative sampling (NEG) [11], and Blackout [12] are included in this category. These approximations train the network to increase the possibilities of positive samples, which are usually labels, and to decrease the probabilities of negative samples, which are randomly sampled. These strategies are beneficial for increasing the training speed. However, their evaluation does not show any improvement in speed.

**Hierarchical softmax** (HS) unifies the softmax function and output vector computation by constructing a tree structure of words. Binary HS [13, 14] uses the binary tree structure, which is $\log(V)$ in depth. However, the binary representation is heavily dependent on each word's position, and therefore, a two-layer [2] or three-layer [15] hierarchy is also introduced. In particular, in the study in [15] several clustered words were arranged in a "short-list," where the outputs of the second level hierarchy were the words themselves, not the classes of the third hierarchy. Adaptive softmax [16] extends the idea and allocates the short-list to the first layer, with a two-layer hierarchy. Adaptive softmax achieves both a training time speedup and a performance gain. HS approaches have advantages for quickly gathering probability of a certain word or predetermined words. However, HS should also visit every word to find the topmost likely words, where the merit of the tree structure is not useful.

**Self-normalization approaches** [17, 18] employ an additional training loss term, which leads a normalization factor $Z$ close to 1. The evaluation of selected words can be achieved significantly faster than by using full-softmax if the denominator is trained well. However, the method cannot ensure that the denominator always appears correctly, and should also consider every word for top-K estimation.

**Differentiated Softmax** (D-softmax) [19] restricts the effective parameters, using the fraction of the full output matrix. The matrix allocates higher dimensional representation to frequent words and only a lower dimensional vector to rare words. From this point of view, there is a commonality between our method and D-softmax in that the length of vector used in the output vector computation varies among words. However, the determination of the length of each portion is somewhat heuristic and requires specified training procedures in D-softmax. The word representation learned

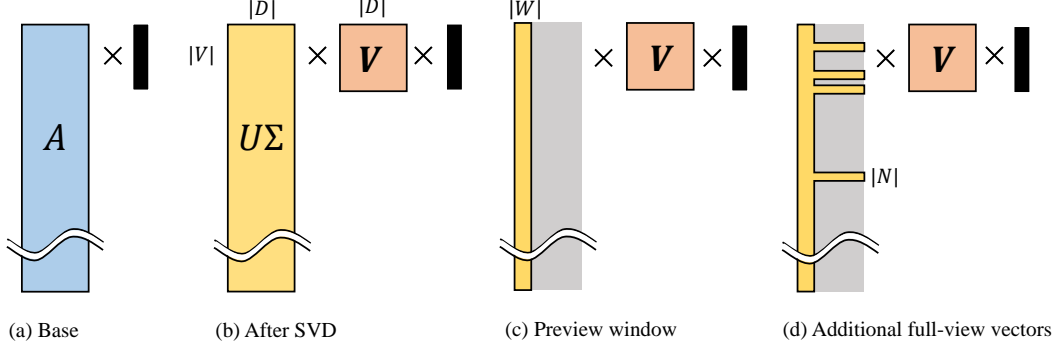

(a) Base      (b) After SVD      (c) Preview window      (d) Additional full-view vectors

Figure 1: Illustration of the proposed SVD-softmax algorithm. The softmax weight matrix is decomposed by singular value decomposition (b). Only a part of the columns is used to compute the preview outputs (c). Selected rows, which are chosen by sorting the preview outputs, are recomputed with full-width (d). For simplicity, the bias vector is omitted.

by D-softmax is restricted from the start, and may therefore be lacking in terms of expressiveness. In contrast, our algorithm first trains words with a full-length vector and dynamically limits the dimension during evaluation. In SVD-softmax, the importance of each word is also dynamically determined during the inference.

## 3 SVD-softmax

The softmax function transforms a $D$-dimensional real-valued vector $\mathbf{h}$ to a $V$-dimensional probability distribution. The probability calculation consists of two stages. First, we acquire the output vector of size $V$, denoted as $z$, from $\mathbf{h}$ by matrix multiplication as

$$\mathbf{z} = \mathbf{Ah} + \mathbf{b} \tag{1}$$

where $\mathbf{A} \in \mathbb{R}^{V \times D}$ is a weight matrix, $\mathbf{h} \in \mathbb{R}^D$ is an input vector, $\mathbf{b} \in \mathbb{R}^V$ is a bias vector, and $\mathbf{z} \in \mathbb{R}^V$ is the computed output vector. Second, we normalize the output vector to compute the probability $y_k$ of each word as

$$y_k = softmax(z_k) = \frac{\exp(A_k h + b_k)}{\sum_{i=1}^{V} \exp(A_i h + b_i)} = \frac{\exp(z_k)}{\sum_{i=1}^{V} \exp(z_i)} = \frac{\exp(z_k)}{Z} \tag{2}$$

The computational complexity of calculating the probability distribution over all classes and only one class is the same, because the normalization factor $Z$ requires every output vector elements to be computed.

### 3.1 Singular value decomposition

SVD is a factorization method that decomposes a matrix into two unitary matrices $\mathbf{U}, \mathbf{V}$ with singular vectors in columns and one diagonal matrix $\mathbf{\Sigma}$ with non-negative real singular values in descending order. SVD is applied to the weight matrix $\mathbf{A}$ as

$$\mathbf{A} = \mathbf{U}\mathbf{\Sigma}\mathbf{V}^T \tag{3}$$

where $\mathbf{U} \in \mathbb{R}^{V \times D}, \mathbf{\Sigma} \in \mathbb{R}^{D \times D}$, and $\mathbf{V} \in \mathbb{R}^{D \times D}$. We multiply $\mathbf{\Sigma}$ and $\mathbf{U}$ to factorize the original matrix into two parts: $\mathbf{U}\mathbf{\Sigma}$ and $\mathbf{V}^{\mathbf{T}}$. Note that $\mathbf{U} \times \mathbf{\Sigma}$ multiplication is negligible in evaluation time because we can keep the result as a single matrix.

Larger singular values in $\mathbf{\Sigma}$ are multiplied to the leftmost columns of $\mathbf{U}$. As a result, the elements of the $\mathbf{B}(= \mathbf{U}\mathbf{\Sigma})$ matrix are statistically arranged in descending order of magnitude, from the first column to the last. The leftmost columns of $\mathbf{B}$ are more influential than the rightmost columns.

---
**Algorithm 1** Algorithm of the proposed SVD-softmax.
---
 1: **input:** trained weight matrix $\mathbf{A}$, input vector $\mathbf{h}$, bias vector $\mathbf{b}$
 2: **hyperparameter:** width of preview window $W$, number of full-view vectors $N$.
 3: **initialize:** decompose $\mathbf{A} = \mathbf{U\Sigma V}^T$, $\mathbf{B} = \mathbf{U\Sigma}$
 4: $\tilde{\mathbf{h}} = \mathbf{V^T} \times \mathbf{h}$
 5: $\tilde{\mathbf{z}} = \mathbf{B}[:,:W] \times \tilde{\mathbf{h}}[:W] + \mathbf{b}$          compute preview outputs with only $W$ dimensions
 6: Sort $\tilde{\mathbf{z}}$ in descending order          select $N$ words of largest preview outputs
 7: $\mathbf{C}_N = \text{Top-}N$ word indices of $\tilde{\mathbf{z}}$
 8: **for** all $id$ in $\mathbf{C}_N$ **do**
 9:     $\tilde{\mathbf{z}}[id] = \mathbf{B}[id,:] \times \tilde{\mathbf{h}} + \mathbf{b}[id]$        update selected words by full-view vector multiplication
10: **end for**
11: $\tilde{Z} = \sum_V \exp \tilde{z_i}$
12: $\tilde{\mathbf{y}} = \exp(\tilde{\mathbf{z}})/\tilde{Z}$          compute probability distribution using softmax
13: return $\tilde{\mathbf{y}}$
---

## 3.2 Softmax approximation

Algorithm 1 shows the softmax approximation procedure, which is also illustrated in Figure 1. Previous methods needed to compare every output vector elements to find the top-K words. Instead of using the full-length vector, we consult every word with a window of restricted length $W$. We call this the "preview window" and the results the "preview outputs." Note that adding the bias $\mathbf{b}$ in preview outputs computation is crucial for the performance. Since larger singular values are multiplied to several leftmost columns, it is reasonable to assume that the most important portion of the output vector is already computed with the preview window.

However, we find that the preview outputs do not suffice to obtain accurate results. To increase the accuracy, $N$ largest candidates $\mathbf{C}_N$ are selected by sorting $V$ preview outputs. The selected candidates are recomputed with the full-length window. We call the candidates the "full-view" vectors. As a result, $N$ outputs are computed exactly while $(V - N)$ outputs are only an approximation based on the preview outputs. In other words, only the selected indices use the full window for output vector computation. Finally, the softmax function is applied to the output vector to normalize the probability distribution. The modified output vector $\tilde{z_k}$ is formulated as

$$\tilde{z_k} = \begin{cases} B_k \tilde{h} + b_k, & \text{if } k \in \mathbf{C}_N \\ B_k[:W]\tilde{h}[:W] + b_k, & \text{otherwise} \end{cases} \tag{4}$$

where $\mathbf{B} \in \mathbb{R}^{V \times D}$ and $\tilde{\mathbf{h}} = V^T \mathbf{h} \in \mathbb{R}^D$. Note that if $k \in \mathbf{C}_N$, $\tilde{z_k}$ is equal to $z_k$. The computational complexity is reduced from $O(V \times D)$ to $O(V \times W + N \times D)$.

## 3.3 Metrics

To observe the accuracy of every word probability, we use Kullback-Leibler divergence ($KLD$) as a metric. $KLD$ shows the closeness of the approximated distribution to the actual one. Perplexity, or negative log-likelihood ($NLL$), is a useful measurement for likelihood estimation. The gap between full-softmax and SVD-softmax $NLL$ should be small. For the evaluation of a given word, the accuracy of probability depends only on the normalization factor $Z$, and therefore we monitor also the denominator of the softmax function.

We define "top-K coverage," which represents how many top-K words of full-softmax are included in the top-K words of SVD-softmax. For the beam-search purpose, it is important to correctly select the top-K words, as beam paths might change if the order is mingled.

## 4 Experimental results

The experiments were performed on three datasets and two different applications: language modeling and machine translation. The WikiText-2 [20] and One Billion Word benchmark (OBW) [21]

Table 1: Effect of the number of hidden units on the WikiText-2 language model. The number of full-view vectors is fixed to 3,300 for the table, which is about 10% of the size of the vocabulary. Top-K denotes top-K coverage defined in 3.3. The values are averaged.

| $D$ | $W$ | $\tilde{Z}/Z$ | $KLD$ | $NLL$ (full/SVD) | Top-10 | Top-100 | Top-1000 |
|---|---|---|---|---|---|---|---|
| 256 | 16 | 0.9813 | 0.03843 | 4.408 / 4.518 | 9.97 | 99.47 | 952.71 |
| | 32 | 0.9914 | 0.01134 | 4.408 / 4.441 | 10.00 | 99.97 | 986.94 |
| 512 | 32 | 0.9906 | 0.01453 | 3.831 / 3.907 | 10.00 | 99.89 | 974.87 |
| | 64 | 0.9951 | 0.00638 | 3.831 / 3.852 | 10.00 | 99.99 | 993.35 |
| 1024 | 64 | 0.9951 | 0.00656 | 3.743 / 3.789 | 10.00 | 99.99 | 992.62 |
| | 128 | 0.9971 | 0.00353 | 3.743 / 3.761 | 10.00 | 100.00 | 998.28 |

datasets were used for language modeling. The neural machine translation (NMT) from German to English was trained with a dataset provided by the OpenNMT toolkit [22].

We first analyzed the extent to which the preview window size $W$ and the number of full-view vectors $N$ affect the overall performance and searched the best working combination.

## 4.1 Effect of the number of hidden units on preview window size

To find the relationship between the preview window's width and the approximation quality, three LMs trained with WikiText-2 were tested. WikiText is a text dataset, which was recently introduced [20]. The WikiText-2 dataset contains 33,278-word vocabulary and approximately 2M training tokens. An RNN with a single LSTM layer [6] was used for language modeling. Traditional full-softmax was used for the output layer. The number of LSTM units was the same as the input embedding dimension. Three models were trained on WikiText-2 with the number of hidden units $D$ being 256, 512, and 1,024.

The models were trained with stochastic gradient descent (SGD) with an initial learning rate of 1.0 and momentum of 0.95. The batch size was set to 20, and the network was unrolled for 35 timesteps. Dropout [23] was applied to the LSTM output with a drop ratio of 0.5. Gradient clipping [24] of maximum norm value 5 was applied.

The preview window widths $W$ selected were 16, 32, 64, and 128 and the number of full-view candidates $N$ were 5% and 10% of the full vocabulary size for all three models. One thousand sequential frames were used for the evaluation. Table 1 shows the results of selected experiments, which indicates that the sufficient preview window size is proportional to the hidden layer dimension $D$. In most cases, $1/8$ of $D$ is an adequate window width, which costs 12.5% of multiplications. Over 99% of the denominator is covered. $KLD$ and $NLL$ show that the approximation produces almost the same results as the original. The top-K words are also computed precisely. We also checked the order of the top-K words that were preserved. The result showed that using too short window width affects the performance badly.

## 4.2 Effect of the vocabulary size on the number of full-view vectors

The OBW dataset was used to analyze the effect of vocabulary size on SVD-softmax. This benchmark is a huge dataset with a 793,472-word vocabulary. The model used 256-dimension word embedding, an LSTM layer of 2,048 units, and a full-softmax output layer. The RNN LM was trained with SGD with an initial learning rate of 1.0.

We explored multiple models by employing a vocabulary size of 8,004, 80,004, 401,951, and 793,472, abbreviated as 8K, 80K, 400K, and 800K below. The 800K model follows the preprocessing consensus, keeping words that appear more than three times. The 400K vocabulary follows the same process as the 800K but without case sensitivity. The 8K and 80K data models were created by choosing the topmost frequent 8K and 80K words, respectively. Because of the limitation of GPU memory, the 800K model was trained with half-precision parameters. We used the full data for training.

Table 2: Effect of the number of full-view vector size $N$ on One Billion Word benchmark language model. The preview window width is fixed to 256 in this table. We omitted the ratio of approximated $\tilde{Z}$ and real $Z$, because the ratio is over 0.997 for all cases in the table. The multiplication ratio is to full-softmax, including the overhead of $\mathbf{V}^T \times \mathbf{h}$.

| $V$ | $N$ | $NLL$ (full/SVD) | Top-10 | Top-50 | Top-100 | Top-500 | Mult. ratio |
|---|---|---|---|---|---|---|---|
| 8K | 1024 | 2.685 / 2.698 | 9.98 | 49.81 | 99.36 | 469.48 | 0.493 |
| | 2048 | 2.685 / 2.687 | 9.99 | 49.99 | 99.89 | 496.05 | 0.605 |
| 80K | 4096 | 3.589 / 3.6051 | 10.00 | 49.94 | 99.85 | 497.73 | 0.195 |
| | 8192 | 3.589 / 3.591 | 10.00 | 49.99 | 99.97 | 499.56 | 0.240 |
| 400K | 16384 | 3.493 / 3.495 | 10.00 | 50.00 | 100.00 | 499.90 | 0.171 |
| | 32768 | 3.493 / 3.495 | 10.00 | 50.00 | 100.00 | 499.98 | 0.201 |
| 800K | 32768 | 4.688 / 4.718 | 10.00 | 49.99 | 99.96 | 499.99 | 0.168 |
| | 65536 | 4.688 / 4.690 | 10.00 | 49.99 | 99.96 | 499.89 | 0.200 |

Table 3: SVD-softmax on machine translation task. The baseline perplexity and BLEU score are 10.57 and 21.98, respectively.

| $W$ | $N$ | Perplexity | BLEU |
|---|---|---|---|
| 200 | 5000 | 10.57 | 21.99 |
| | 2500 | 10.57 | 21.99 |
| | 1000 | 10.58 | 22.00 |
| 100 | 5000 | 10.58 | 22.00 |
| | 2500 | 10.59 | 22.00 |
| | 1000 | 10.65 | 22.01 |
| 50 | 5000 | 10.60 | 22.00 |
| | 2500 | 10.68 | 21.99 |
| | 1000 | 11.04 | 22.00 |

The preview window width and the number of full-view vectors were selected in the powers of 2. The results were computed on randomly selected 2,000 consecutive frames.

Table 2 shows the experimental results. With a fixed hidden dimension of 2,048, the required preview window width does not change significantly, which is consistent with the observations in Section 4.1. However, the number of full-view vectors $N$ should increase as the vocabulary size grows. In our experiments, using 5% to 10% of the total vocabulary size as candidates sufficed to achieve a successful approximation. The results prove that the proposed method is scalable and more efficient when applied to large vocabulary softmax.

### 4.3 Result on machine translation

NMT is based on neural networks and contains an internal softmax function. We applied SVD-softmax to a German to English NMT task to evaluate the actual performance of the proposed algorithm.

The baseline network, which employs the encoder-decoder model with an attention mechanism [25, 26], was trained using the OpenNMT toolkit. The network was trained with concatenated data which contained a WMT 2015 translation task [27], Europarl v7 [28], common crawl [29], and news commentary v10 [30], and evaluated with newstest 2013. The training and evaluation data were tokenized and preprocessed by following the procedures in previous studies [31, 32] to conduct case-sensitive translation with 50,004 frequent words. The baseline network employed 500-dimension word embedding, encoder- and decoder-networks with two unidirectional LSTM layers with 500 units each, and a full-softmax output layer. The network was trained with SGD with an initial learning rate of 1.0 while applying dropout [23] with ratio 0.3 between adjacent LSTM layers. The rest of the training settings followed the OpenNMT training recipe, which is based on

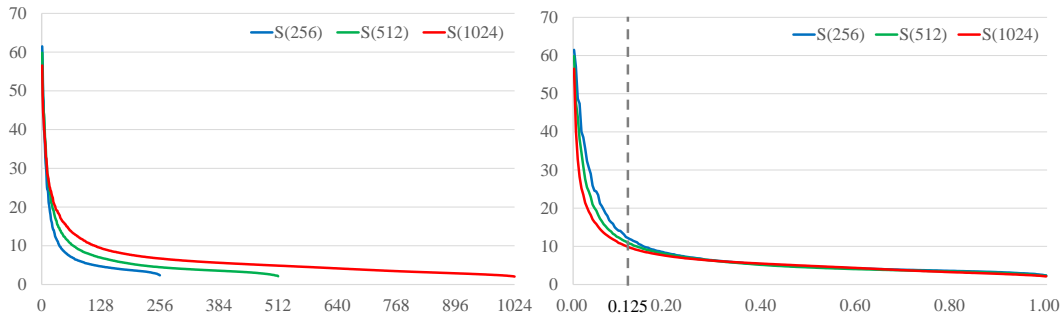

Figure 2: Singular value plot of three WikiText-2 language models that differ in hidden vector dimension $D \in \{256, 512, 1024\}$. The left hand side figure represents the singular value for each element, while the right hand side figure illustrates the value proportional to $D$. The dashed line implies $0.125 = 1/8$ point. Both are from the same data.

previous studies [31, 33]. The performance of the network was evaluated according to perplexity and the case-sensitive BLEU score [34], which was computed with the Moses toolkit [35]. During translation, a beam search was conducted with beam width 5.

To evaluate our algorithm, the preview window widths $W$ selected were 25, 50, 100, and 200, and the numbers of full-view candidates $N$ chosen were 1,000, 2,500, and 5,000.

Table 3 shows the experimental results for perplexity and the BLEU score with respect to the preview window dimension $W$ and the number of full-view vectors $N$. The full-softmax layer in the baseline model employed a hidden dimension $D$ of 500 and computed the probability for $V = 50,004$ words. The experimental results show that a speed up can be achieved with preview width $W = 100$, which is $1/5$ of $D$, and the number of full-view vectors $N = 2,500$ or 5,000, which is $1/5$ or $1/10$ of $V$. The parameters chosen did not affect the translation performance in terms of perplexity. For a wider $W$, it is possible to use a smaller $N$. The experimental results show that SVD-softmax is also effective when applied to NMT tasks.

## 5 Discussion

In this section, we provide empirical evidence of the reasons why SVD-softmax operates efficiently. We also present the results of an implementation on a GPU.

### 5.1 Analysis of $W$, $N$, and $D$

We first explain the reason the required preview window width $W$ is proportional to the hidden vector size $D$. Figure 2 shows the singular value distribution of WikiText-2 LM softmax weights. We observed that the distributions are similar for all three cases when the singular value indices are scaled with $D$. Thus, it is important to preserve the ratio between $W$ and $D$. The ratio of singular values in a $D/8$ window over the total sum of singular values for 256, 512, and 1,024 hidden vector dimensions is 0.42, 0.38, and 0.34, respectively.

Furthermore, we explore the manner in which $W$ and $N$ affect the normalization term, i.e., the denominator. Figure 3 shows how the denominator is approximated while changing $W$ or $N$. Note that the leftmost column of Figure 3 represents that no full-view vectors were used.

### 5.2 Computational efficiency

The modeled number of multiplications in Table 2 shows that the computation required can be decreased to 20%. After factorization, the overhead of matrix multiplication $\mathbf{V}^T$, which is $O(D^2)$, is a fixed cost. In most cases, especially with a very large vocabulary, $V$ is significantly larger than $D$, and the additional computation cost is negligible. However, as $V$ decreases, the portion of the overhead increases.

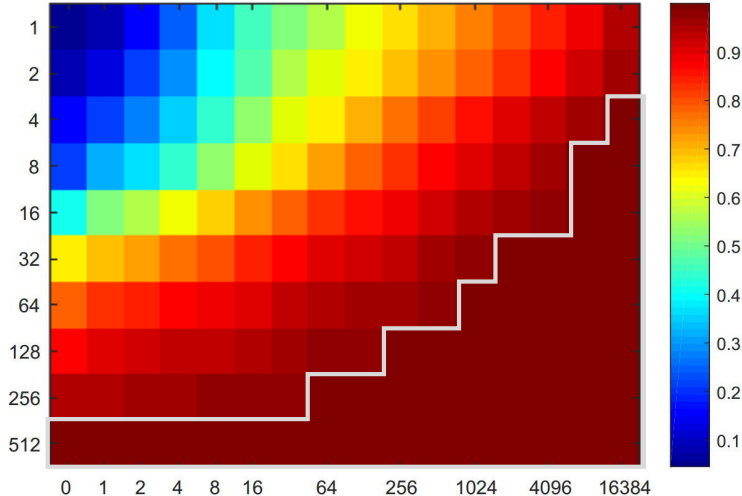

Figure 3: Heatmap of approximated normalization factor ratio $\tilde{Z}/Z$. The x and y axis represent $N$ and $W$, respectively. The WikiText-2 language model with $D = 1{,}024$ was used. Note that the maximum values of $N$ and $W$ are 1,024 and 33,278, respectively. The gray line separates the area by 0.99 as a threshold. Best viewed in color.

Table 4: Measured time (ms) of full-softmax and SVD-softmax on a GPU and CPU. The experiment was conducted on a NVIDIA GTX Titan-X (Pascal) GPU and Intel i7-6850 CPU. The second column indicates the full-softmax, while the other columns represent each step of SVD-softmax. The cost of the sorting, exponential, and sum is omitted, as their time consumption is negligible.

| | Full-softmax | SVD-softmax | | | |
|---|---|---|---|---|---|
| | $\mathbf{A} \times \mathbf{h}$ | $\mathbf{V}^T \times \mathbf{h}$ | Preview window | Full-view vectors | Sum (speedup) |
| Device | $(262k, 2k)$ $\times 2k$ | $(2k, 2k)$ $\times 2k$ | $(262k, 256)$ $\times 256$ | $(16k, 2k)$ $\times 2k$ | - |
| GPU | 14.12 | 0.33 | 2.98 | 1.12 | 4.43 ($\times 3.19$) |
| CPU | 1541.43 | 25.32 | 189.27 | 88.98 | 303.57 ($\times 5.08$) |

We provide an example of time consumption on a CPU and GPU. Assume the weight $A$ is a 262K ($V = 2^{18}$) by 2K ($D = 2^{11}$) matrix and SVD-softmax is applied with preview window width of 256 and the number of full-view vectors is 16K ($N = 2^{14}$). This corresponds to $W/D = 1/8$ and $N/V = 1/16$. The setting well simulates the real LM environment and the use of the recommended SVD-softmax hyperparameters discussed above. We used our highly optimized custom CUDA kernel for the GPU evaluation. The matrix $\mathbf{B}$ was stored in row-major order for convenient full-view vector evaluation.

As observed in Table 4, the time consumption is reduced by approximately 70% on the GPU and approximately 80% on the CPU. Note that the GPU kernel is fully parallelized while the CPU code employs a sequential logic. We also tested various vocabulary sizes and hidden dimensions on the custom kernel, where a speedup is mostly observed, although it is less effective for small vocabulary cases.

## 5.3 Compatibility with other methods

The proposed method is compatible with a neural network trained with sampling-based softmax approximations. SVD-softmax is also applicable to hierarchical softmax and adaptive softmax, especially when the vocabulary is large. Hierarchical methods need large weight matrix multiplication to gather every word probability, and SVD-softmax can reduce the computation. We tested SVD-softmax with various softmax approximations and observed that a significant amount of multiplication is removed while the performance is not significantly affected as it is by full softmax.

# 6 Conclusion

We present SVD-softmax, an efficient softmax approximation algorithm, which is effective for computing top-K word probabilities. The proposed method factorizes the matrix by SVD, and only part of the SVD transformed matrix is previewed to determine which words are worth preserving. The guideline for hyperparameter selection was given empirically. Language modeling and NMT experiments were conducted. Our method reduces the number of multiplication operations to only 20% of that of the full-softmax with little performance degradation. The proposed SVD-softmax is a simple yet powerful computation reduction technique.

### Acknowledgments

This work was supported in part by the Brain Korea 21 Plus Project and the National Research Foundation of Korea (NRF) grant funded by the Korea government (MSIP) (No.2015R1A2A1A10056051).

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
