[Reviews · NeurIPS 2017]

Reviewer 1



This paper proposes an efficient way to approximate the softmax computation for large vocabulary applications. The idea is to decompose the output matrix with singular value decomposition. Then, by selecting the most important singular values you select the most probable words and also compute the partition function for a limited amount of words. These are supposed to contribute for the most part of the sum. For the remaining words, their contributions to the partition function is only approximated. The idea is nice and the experimental results show a significant inference speed up, along with similar performance. My main concern is that sometimes the paper is poorly written. Just one example: the authors use the term "logit" to denote the exponential of a dot product. This is not correct. The logit function is the inverse of the sigmoidal "logistic" function. This choice of term is therefore incorrect and misleading. More examples are given below. However the idea is both simple and nice. - Page 2, line 73: Moreover, self normalized models requires a huge training time. - Page 3, line 93: It is not "approximately". It's the same. The softmax function outputs the probability distribution over the whole vocabulary. - Line 102: reword this sentence - Line 114: for obtaining accurate results -> to obtain accurate results - Exactly instead of correctly in line 117 Page 5: Why you didn't use the full softmax on the wiki data ? The vocabulary is small and the computational cost is really cheap! Page 6: The script to compute the BLEU score is included in the Moses released, but it's not the Moses toolkit. However, which one did you use ? To have a stronger baseline you could use bpe to decompose words. Moreover, it could be more meaningful to have a larger vocabulary. For instance if you translate from English to German, the output vocabulary is really larger, even if you use bpe. I understand the goal is to show that your svd softmax performs as well as the full softmax with a huge gain in time. However, this gain of time could benefit to the NMT system by allowing a longer training in a state of the art setup with the svd softmax.

Reviewer 2



This paper presents a simple approach to speed-up the softmax computation for large vocabulary neural networks. The perform SVD on the output embedding matrix. At inference time, they first compute N-top candidates (preview logits) based only on W dimensions where W < embedding dimension, and then compute a softmax over the entire vocabulary but computing the actual values of the logits corresponding to the top N candidates. This reduces the complexity from O(VD) to O(VW + ND). They evaluate their approach using three metrics 1. KL divergence between the approximate and actual softmax 2. Perplexity 3. top k coverage since that is important for beam search Their results show that on language modeling tasks and the 1 billion word benchmark, their approach seems to perform comparable with using full softmax with a reduction of 70% complexity on the GPU. This is a simple approach and it seems to work. It would have been good to see if it stands up when tried on SOTA systems for MT and language modeling.

Reviewer 3



This paper addresses the question how to efficiently EVALUATE the softmax at the output layer of an RNN/LSTM. The underlying idea is to use an SVD decomposition of the output layer. The authors first present results for LM on the One billion word benchmark. Their goal is not to measure perplexity, but to be able to know the top K LM probabilities (with correct normalization). I'm not impressed by the results in Table 2. In the best case, the proposed approach is about 6x faster than the full softmax over the 800k vocabulary. This still corresponds to an 130k softmax which is quite expensive. There are alternative approaches like byte pair encoding (which is very successfully used in NMT), and which use vocabularies in the range of 20-30k. Also, the authors clearly state that their focus is on fast top K evaluation, but they still need to train the model. I consider that training a full 800k softmax is prohibitive ! The authors admit themselves that they had to use half-precision parameters to fit the model into GPU memory. Of course, one could use NCE or similar techniques during training, but the authors should then show that their method is compatible with NCE training. In my opinion, an efficient softmax speed-up technique should consider training and evaluation ! The authors then provide results for NMT: the proposed method has no impact on the BLEU score. Finally, it's nice that the authors have an optimized implementation on GPU. Do they plan to make it freely available ? Overall, they achieve a 3x speed improvement on GPU in comparison to a full 260k vocabulary. As stated above, training such huge vocabularies is prohibitive. In my opinion, the real question is whether the proposed method can be used with (BPE) vocabularies of 20-64k and still give a measured 3-5x improvement in evaluation mode on GPU. That would be a much better message. I also would like to see comparative results (performance and speed) with respect to other softmax speed-up techniques. You need to show that your method is better than other speed-up techniques, not in comparison to a huge softmax (which is never used in practice ...)